# Increase trajectories of tendon micro vibration intensity during ankle plantar flexion: A longitudinal data analysis using latent curve models

**Tatsuhiko Matsumoto**[1,2]*, **Yutaka Kano**[1‡]

**1** Graduate School of Engineering Science, Osaka University, Toyonaka, Osaka, Japan, **2** Murata Manufacturing Co., Ltd., Kyoto, Japan

‡ Current address: Faculty of Culture and Information Science, Doshisha University, Kyotanabe, Kyoto, Japan

* u960547e@ecs.osaka-u.ac.jp

**Data availability statement:** The anonymized and processed dataset generated and analyzed

## Abstract

We focus on fine vibrations originating from tendons (Mechanotendography: MTG) as a novel method for quantifying muscle activity. Quantifying muscle activity using MTG can enable daily and long-term continuous measurements, which have been challenging for electromyography (EMG) and mechanomyography (MMG). However, the detailed trajectory of MTG increase relative to exerted muscle strength has not been clarified, nor has the mechanism of MTG generation. Our research has two objectives. The first is to clarify the detailed relationship between exerted muscle strength levels and MTG through statistical modeling. The second is to establish a highly accurate hypothesis concerning the mechanism of MTG generation based on the modeling results and physiological knowledge. We focused on the Achilles tendon to study these two objectives. Experiments were conducted on 62 participants, and MTG data were obtained at various levels of exerted muscle strength. The obtained data were structured into a longitudinal data format representing the trajectory of MTG increase with increasing muscle strength. We used latent curve models (LCM) to identify this structure. By applying various LCMs to explore an optimal model, we found that the quadratic LCM received the best fit for females, while the piecewise linear LCM with a breakpoint at 50% exerted muscle strength received the best fit for males. Notably, a significant sex difference was observed in the rate of increase in MTG at low levels of exerted muscle strength. These results suggest that MTG is caused by fine vibrations generated by muscle fiber contractions, and these fine vibrations are transmitted to the tendons connected to the muscles, where they are observed. Future research will focus on verifying this hypothesis through increased time points and physiological experiments.

during the current study is available on Figshare: https://doi.org/10.6084/m9.figshare.29123906.v1. Due to ethical considerations, raw data cannot be made publicly available. However, requests for access to the minimal data can be directed to the Ethics Committee of Shikoku Medical School at c.fujisawa@459.ac.jp, who will review requests in accordance with institutional policy. This ensures long-term and stable access to data beyond the authors' tenure.

**Funding:** The author(s) received no specific funding for this work.

**Competing interests:** The authors have declared that no competing interests exist.

## Introduction

Quantifying muscle activity is essential for optimizing exercise and rehabilitation programs [1,2]. Based on such insights, applications have also been developed to enhance the efficiency of rehabilitation and exercise [3,4]. Furthermore, not only real-time observation but also long-term and continuous quantification can enable the improvement and optimization of plans [5].

Traditionally, electromyography (EMG) and mechanomyography (MMG) have been used as methods to quantify muscle activity [6,7]. However, these methods have several issues. The first issue is that their measurement requires specialized knowledge. EMG and MMG require the placement of electrodes or microphones on the muscle belly, which demands specialized skills, making it difficult for general users to measure in daily life. The second issue is that the electrodes or microphones must be adhered to the skin surface. Due to sweat or friction with clothing, they can peel off or cause skin problems, making long-term measurement difficult.

To address the issues related to the long-term quantification of muscle activity, we focused on biological signals originating from tendons, which differ from EMG and MMG. It is known that muscles emit biological signals that contribute to the quantification of muscle activity, such as MMG and physiological tremors derived from nervous system reflexes [8]. Since tendons are connected to muscles, capturing such biological signals transmitted to the tendons could potentially quantify muscle activity. Schaefer et al. defined the biological signals emitted from tendons as mechanotendography (MTG) [9]. Capturing biological signals from tendons could solve the issues EMG and MMG face. Tendons can be easily palpated without special skills, are less affected by clothing friction, and contain less subcutaneous fat—factors that support stable, long-term signal acquisition. These factors reduce the burden of long-term measurement.

We focused on the Achilles tendon as the tendon to verify these benefits and developed an anklet-type device. The Achilles tendon is connected to multiple calf muscles. It was anticipated that biological signals from multiple muscles would be superimposed and transmitted to the tendon, capturing high-intensity MTG. Additionally, as mentioned earlier, the area between the skin and the tendon has less fat, which could attenuate signals, reducing individual differences. This device uses a piezoelectric film sensor to capture signals. In our previous research, we successfully captured MTG that monotonically increased as did exerted muscle strength [10]. However, the detailed trajectory of MTG increase relative to exerted muscle strength has not been clarified. The mechanism of MTG generation has not been addressed, either.

This study aims to (1) statistically model the relationship between MTG and exerted muscle strength, and (2) propose a hypothesis on the mechanism of MTG generation based on physiological interpretation. We collected data from 62 participants using a data acquisition system to achieve these research objectives. Participants performed isometric ankle plantar flexion exercises at five levels of exerted muscle strength. By extracting sensor features for the five levels of exerted muscle strength, we processed the data into a longitudinal structure where each 62 individuals had data at five different points in time. We then applied latent curve models (LCM), defined within the structural equation modeling (SEM) framework, to this longitudinal data. We varied the structure of the LCMs based on hypotheses. We evaluated their goodness-of-fit, identifying the model structure that best represents the process of MTG amplification as the exerted muscle strength increases. Additionally, we derived hypotheses on the mechanism of MTG generation by examining the commonalities between the model structures and physiological knowledge.

## Materials and methods

### Data acquisition experiments

**Participants.** This experiment involved 62 healthy participants (30 males and 32 females) aged between 21 and 58. The experimental protocol was designed in accordance with the Helsinki Declaration, and written informed consent was obtained from all participants before the experiment. The study was approved by the Ethics Committee of Shikoku Medical School (approval numbers: R03-03-003, R03-03-004), and the recruitment period was from September 16, 2020, to March 31, 2021. Additionally, approval for secondary use of the experimental data was obtained from the same Ethics Committee (approval numbers: R05-08-001, R05-08-002).

Participants in this study were selected based on the following two criteria:

- Individuals aged 20 to 64 who are capable of making their own decisions.
- Individuals who understood the purpose of the study and agreed to participate.

Additionally, individuals meeting the following conditions were excluded:

- Minors under the age of 20.
- Individuals receiving treatment for existing diseases.
- Individuals monitoring the course of existing diseases as part of medical care.
- Individuals deemed unsuitable for the study by the principal investigator.

The sample size was chosen to recruit as many participants as possible to ensure the reliability of the analytical results. Participant recruitment was conducted through specific institutions with established cooperation relationships and individuals associated with these institutions were invited to participate. Basic physical information deemed relevant to the muscle condition was measured before the experiment.

**Data acquisition system.** We developed a system to acquire MTG from the Achilles tendon via the skin surface near the tendon. The system is illustrated in Fig 1. For the MTG acquisition, we used a biodegradable piezoelectric film sensor (Picoleaf®, Murata Manufacturing Co., Ltd.) [11]. This sensor exhibits non-pyroelectric characteristics, making it less susceptible to temperature changes such as body heat. The piezoelectric film sensor generates an electric charge in response to shear stress on the material, allowing it to detect minute strain changes with high sensitivity. This capability enables the detection of weak vibrations, such as MTG, which are attenuated when they reach the skin surface. Additionally, this sensor is a thin, flexible film. Therefore, it can be deformed to fit the shape of the human body, allowing for efficient acquisition of signals from the skin surface [12]. The sensor's sampling frequency was set to 1000 Hz in this system. The sensor was positioned to cover the Achilles tendon at the ankle.

**Exercise protocol.** Participants performed isometric ankle plantar flexion exercises while the developed device was attached, and we acquired data during these exercises. Potential noise in this experiment can be classified as follows:

- Experimental noise:
    Noise arising from the experimental setup, such as body movement, system positioning, or participants not performing as intended during the experiment.
- Physiological noise:
    Noise is caused by the participants' attributes, such as day-to-day differences and individual differences in physical and neurological characteristics.

Before installation    After installation

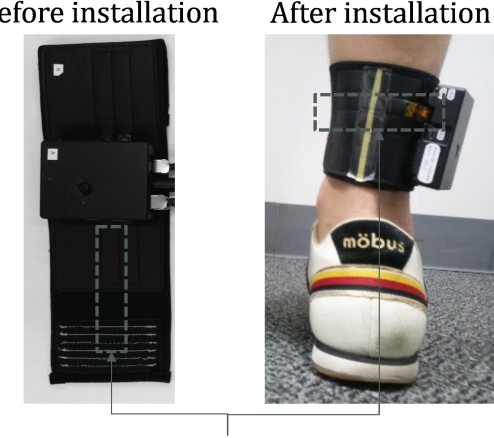

piezoelectric film sensor

**Fig 1. Description of system components and placement of the device.** The device is attached around the ankle. The piezoelectric film sensor embedded in the device is positioned to fit the Achilles tendon during this process. A smartphone connected via Bluetooth can be used to control the device's start and end of measurements.

The experimental protocol was designed to minimize the experimental noise as much as possible. To suppress the experimental noise due to body movement and system positioning, a Cybex isokinetic dynamometer (Cybex NORM®, HUMAC, CA, USA) was used as a performance evaluation device [13]. Fig 2 shows a participant fixed in the Cybex device. Cybex was calibrated according to the manufacturer's recommendations before the clinical trials. The seat position was adjusted so that the axis of rotation of the participant's knee (the lateral epicondyle of the femur) aligned with the mechanical axis of the dynamometer. The knee angle was fixed at 90 degrees. The shin pad was placed just above the lateral malleolus, and the ankle angle was fixed at 90 degrees. Data were collected from the right foot regardless of the participant's dominant foot or hand.

The experiment consisted of three phases to reduce experimental and physiological noise: warm-up, practice, and data acquisition. The purpose of the warm-up phase was to homogenize the subjects' different physical and neurological characteristics, thereby reducing physiological noise. Another aim of this phase was to ensure that no abnormalities occurred in the subjects' bodies during the experiment. In the practice phase, the experimenter explained the operation to the subjects and provided sufficient practice time (about 1 min). This was to reduce experimental noise caused by participants not performing as intended. After these phases were completed, the data acquisition phase began. Fig 3 shows the procedure for the data acquisition phase. Here, $i$ ($i = 1, \ldots, 5$) represents the exertion force number, and $j$ ($j = 1, \ldots, 62$) represents the participant ID. $p_{ij}[\%]$ ($p_{ij} = 0, 25, 50, 75, 100$) indicates the percentage of maximum voluntary contraction (MVC) exerted by each participant. $F_p[N \cdot m]$ is the torque value that each participant is required to maintain during the data acquisition at $p_{ij}$. The Cybex device includes a dynamometer, which measures the torque values. Additionally, the Cybex device is equipped with a real-time monitor that displays the torque values measured by the dynamometer. Participants use this monitor to maintain and control their exerted force. The maximum torque value measured at $p_{ij} = 100$ is defined as $F_{100}$. The set torque values for $p_{ij} = 25, 50, 75$ are automatically determined based on $F_{100}$ as follows: $F_{25} = 0.25F_{100}$, $F_{50} = 0.5F_{100}$, and $F_{75} = 0.75F_{100}$.

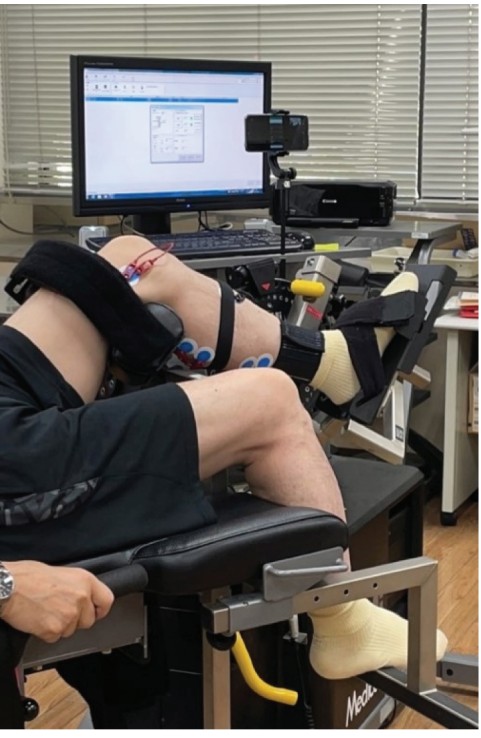

**Fig 2. The appearance of the subject secured in the Cybex machine.** Pushing the ankle towards the right side of the image is called plantar flexion. Participants perform plantar flexion exercises. A pedal is placed under the foot, which allows the force exerted on the pedal to be measured as torque. The monitor can display the current torque value in real-time. Additionally, it is possible to draw a line at any target torque value.

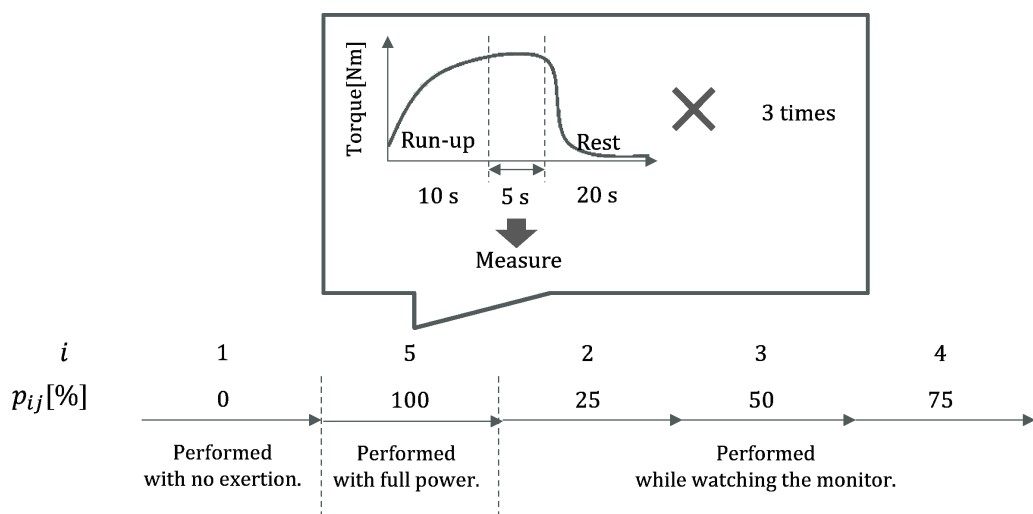

**Fig 3. Data acquisition procedure. Participants gradually increase their exertion over 10 sec and then maintain the specified torque value for 5 sec.** The measurement begins when the participant starts maintaining the specified torque value. This procedure is repeated three times for each output level.

Since the set torque values need to be determined based on $F_{100}$, each participant performed the isometric ankle plantar flexion exercises in the order of $p_{ij} = 0, 100, 25, 50, 75$. The data obtained under the condition of $p_{ij} = 0$ were defined as the data obtained when the participant was fixed in the Cybex device without exerting any force. The data obtained under the condition of $p_{ij} = 100$ were defined as the data obtained when the participant was instructed to perform the isometric ankle plantar flexion exercise with maximum effort. The data obtained under the conditions of $p_{ij} = 25, 50, 75$ were defined as the data obtained when the participant was instructed to maintain the set torque value while performing the isometric ankle plantar flexion exercise.

Each $p_{ij}$ experiment involved three sets of 5-second data acquisition. Participants were instructed to gradually increase their muscle strength over 10 sec to reach the set torque value. In particular, the examiner monitored the display for the $p_{ij} = 25, 50, 75$ experiments and confirmed that the exerted muscle strength had reached the set torque value before starting the measurement. A 20-second rest period was provided between each set. Fig 4 shows each participant's box plot of physical information and $F_{100}$ for each participant.

## Data pre-processing

As described in section Exercise protocol, while the potential for significant, consistent experimental noise such as dynamic movements was minimized, small, consistent biological noise and irregular experimental noise could not be completely eliminated. To accurately capture and quantify the magnitude of the primary vibrations, we preprocessed the data by performing noise removal and peak detection before calculating the RMS. Noise removal and peak detection in the preprocessing were performed using Python version 3.8.8 with the Scipy library version 1.10.0 [14].

**Noise removal.** To mitigate the influence of noise and extract only the main vibration components, a moving average filter and a Gaussian filter were applied [15,16]. First, data smoothing was performed using a moving average filter. The moving average filter reduces irregular and abrupt noise by calculating the simple average of data points within a window. Let $t$ $(1 \leq t \leq T_{ij})$ be the time when the raw data from the piezoelectric film sensor was acquired, and let $s_{ij}(t)$ be the value of the piezoelectric film sensor at time $t$. The value of the piezoelectric film sensor after processing with the moving average filter, $s'_{ij}(t)$, can be

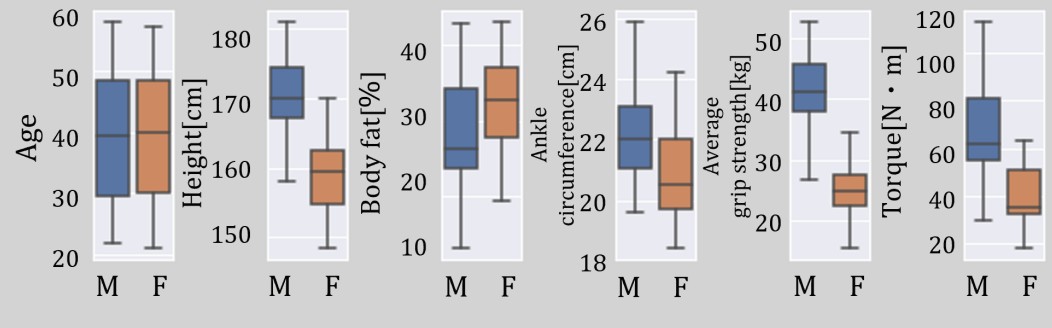

**Fig 4. Box plot of participant's physical information by sex.** The top of the box represents the third quartile, the bottom of the box represents the first quartile, and the horizontal line in the middle of the box represents the median. The top whisker represents the maximum value, and the bottom whisker represents the minimum value.

expressed as

$$s'_{ij}(t) = \frac{1}{N} \sum_{k=0}^{N-1} s_{ij}(t + \frac{k}{1000}).$$ (1)

In this study, a moving average filter with a window width of $N = 20$ ms was applied.

The Gaussian filter was further applied to the smoothed data $s'_{ij}(t)$ obtained from the moving average filter. The Gaussian filter filters the surrounding data points based on the Gaussian function. This effectively removes small, consistent noise of specific frequencies. Additionally, the Gaussian filter applies a Gaussian function to each data point and calculates a weighted average with neighboring data points. This process effectively removes noise while preserving the overall trends and tendencies of the data. The value of the piezoelectric film sensor after processing with the Gaussian filter, $s''_{ij}(t)$, can be expressed as

$$s''_{ij}(t) = \sum_{k=-r}^{r} s'_{ij}(t + \frac{k}{1000}) \frac{1}{\sqrt{2\pi\sigma^2}} e^{-\frac{k^2}{2\sigma^2}}.$$ (2)

The standard deviation of the Gaussian filter was set to $\sigma = 5$. The kernel radius was set to $r = 4\sigma$, resulting in a kernel size of 41, including the central point. To handle data at the endpoints, the input was extended by reflecting it about the edge of the last pixel, a technique also known as half-sample symmetric reflection.

**Peak detection.** The piezoelectric film sensor generates an electric charge in response to the magnitude of shear stress. This means the sensor outputs the magnitude of acceleration when it bends. Peak points represent the maximum acceleration when the sensor bends. Previous studies using the same sensor have confirmed that the amplitude increases as the exerted muscle strength increases [10]. Therefore, peak points can be considered to represent the magnitude of the primary vibrations. On the other hand, points other than peaks represent the intermediate state of the sensor bending. Points other than peaks can become noise when quantifying, as the number of time points is influenced by the time taken for the sensor to deform. Therefore, peak points were extracted. Peaks coincide with stationary points since noise has already been removed in section Noise removal. The index set of stationary points $v$ can be expressed as

$$v = \left\{ t \mid \left( \frac{d}{dt} s''_{ij}(t) \cdot \frac{d}{dt} s''_{ij}(t-1) \right) < 0 \right\},$$ (3)

and the set of stationary points can be expressed as

$$S_{ij}(t) = \{ s''_{ij}(t) \mid t \in v \}.$$ (4)

**Calculation of RMS value.** The root mean square (RMS) is used as a feature to quantify the amplitude [17]. The RMS value $Y_{ij}$ for each exerted muscle strength of each participant is defined as

$$Y_{ij} = \sqrt{\frac{1}{T_{ij}} \sum_{t=1}^{T_{ij}} (S_{ij}(t))^2}.$$ (5)

In this study, this RMS value is treated as the observed value. By extracting the RMS values for the five levels of exerted muscle strength, the data structure becomes longitudinal data with 62 individuals having five observation points each, as shown in Fig 5.

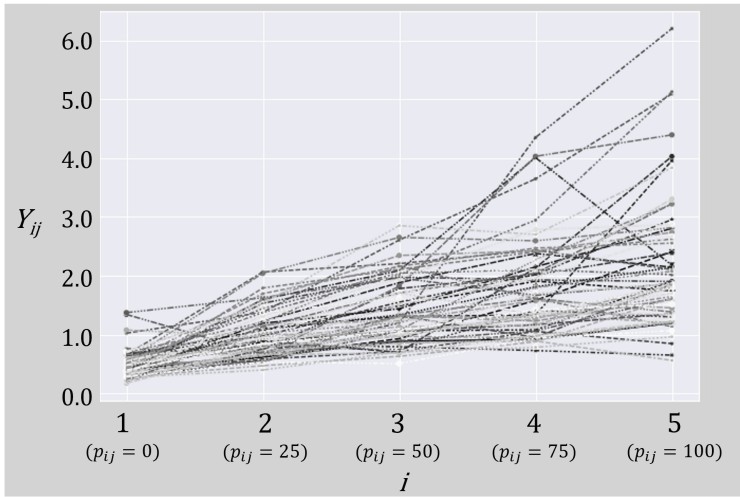

**Fig 5. Trajectory of the five observation points for each participant.**

## SEM methods

In this study, we base our analysis on the LCM, which is well-suited for longitudinal data. LCM is a type of model handled within the framework of SEM and is a statistical model for modeling longitudinal phenomena [18]. LCM can estimate not only the trajectory of within-individual changes over time but also the individual differences in changes between individuals. In other words, it is a model that can simultaneously estimate parameters associated with general group trends that change over time and how these parameters vary among individuals within the population.

The basic procedure of LCM involves selecting the optimal model while examining the information criteria and goodness-of-fit for each model applied to the data and identifying the true hypotheses. The statistics used to evaluate goodness-of-fit can confirm not only the significance of parameters but also the overall absolute fit of the model. In this study, we chose to check the following five measures of fit: Root Mean Square Error of Approximation (RMSEA), Standardized Root Mean Square Residual (SRMR), Comparative Fit Index (CFI), Tucker-Lewis Index (TLI), and Adjusted Goodness of Fit Index (AGFI). It is generally indicated that CFI >0.9, TLI >0.9, and AGFI >0.9 suggest a good model fit [19,20]. Similarly, RMSEA <0.08 and SRMR <0.08 are considered good, whereas RMSEA >0.1 and SRMR >0.1 indicate poor fit [21,22]. TLI, CFI, and AGFI are indicators focused on whether the proposed model fits the observed data overall. On the other hand, RMSEA and SRMR consider the differences, uncertainties, and error rates between the actual observed data and the predicted results. Since different indicators define fit differently, we report all these indicators. For the analysis of structural equation modeling (SEM), R version 4.22 with the lavaan package version 0.6.16 was used [23].

**Linear LCM.** The linear latent growth model can be expressed as

$$Y_j = \Lambda \eta_j + \epsilon_j, \ j = 1, \dots, n. \tag{6}$$

Here, $Y_j$ is an observable random vector for participant $j$, represented as

$$Y_j = \begin{bmatrix} Y_{1j} \\ Y_{2j} \\ Y_{3j} \\ Y_{4j} \\ Y_{5j} \end{bmatrix} \tag{7}$$

and $\mathbf{\Lambda}$ is a factor loading matrix in a linear LCM, it is represented as

$$\mathbf{\Lambda} = \begin{bmatrix} \lambda_{11} & \lambda_{12} \\ \vdots & \vdots \\ \lambda_{51} & \lambda_{52} \end{bmatrix} = \begin{bmatrix} 1 & 0 \\ 1 & 1 \\ 1 & 2 \\ 1 & 3 \\ 1 & 4 \end{bmatrix}. \tag{8}$$

The latent vector $\eta_j$ consists of $\eta_{1j}$ and $\eta_{2j}$ representing an intercept and trend of a fitted line for participant $j$. For these random variables, suppose the following conditions:

$$\eta_j \sim \mathrm{MVN}(\boldsymbol{\alpha}, \mathbf{\Phi}), \tag{9}$$

$$E[\eta_j] = \boldsymbol{\alpha} = \begin{bmatrix} \alpha_1 \\ \alpha_2 \end{bmatrix}, \tag{10}$$

$$\mathrm{Var}[\eta_j] = \mathbf{\Phi} = \begin{bmatrix} \phi_{11} & \phi_{12} \\ \phi_{21} & \phi_{22} \end{bmatrix}. \tag{11}$$

$\epsilon_j$ is the error term vector for participant $j$, expressed as

$$\epsilon_j = \begin{bmatrix} \epsilon_{1j} \\ \vdots \\ \epsilon_{5j} \end{bmatrix}. \tag{12}$$

The random vector $\epsilon_j$ follows according to a multivariate normal distribution with mean vector $\mathbf{0}$ and variance-covariance matrix $\mathbf{\Theta}$, as follows:

$$\epsilon_j \sim \mathrm{MVN}(\mathbf{0}, \mathbf{\Theta}) \tag{13}$$

$$E[\epsilon_j] = \mathbf{0} \tag{14}$$

$$\mathrm{Var}[\epsilon_j] = \mathbf{\Theta}. \tag{15}$$

In this study, we assume that the error variances can differ at each time point and that there is no correlation between errors at different time points. As a result, we have

$$\mathrm{Var}[\epsilon_j] = \mathbf{\Theta} = \begin{bmatrix} \theta_1 & 0 & 0 & 0 & 0 \\ 0 & \theta_2 & 0 & 0 & 0 \\ 0 & 0 & \theta_3 & 0 & 0 \\ 0 & 0 & 0 & \theta_4 & 0 \\ 0 & 0 & 0 & 0 & \theta_5 \end{bmatrix}. \tag{16}$$

Fig 6 shows the path diagram of the linear LCM.

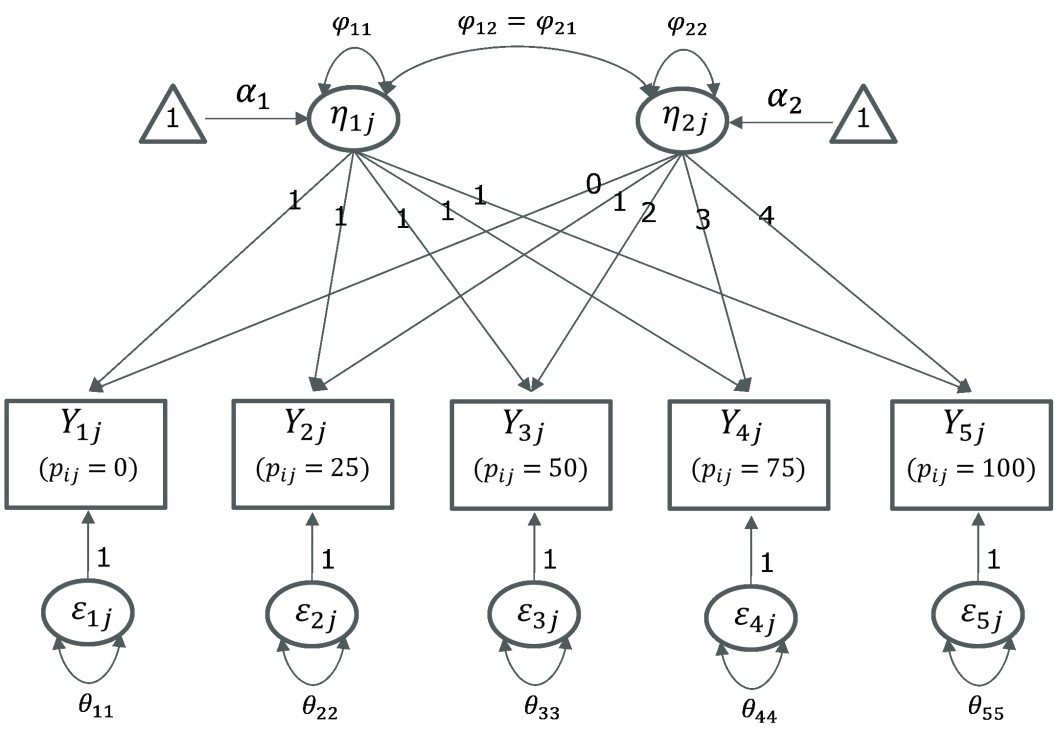

**Fig 6. Path diagram of the linear LCM.**

**Quadratic LCM.** To account for cases where there is no clear changing point and the slope changes smoothly, we can apply a quadratic LCM. Unlike the linear LCM, the latent variable vector for participant $j$ is

$$\boldsymbol{\eta}_j = \begin{bmatrix} \eta_{1j} \\ \eta_{2j} \\ \eta_{3j} \end{bmatrix}, \tag{17}$$

where an additional latent variable related to the quadratic slope component is included. Since $\boldsymbol{\eta}_j$ remains a vector of random variables, the following conditions hold:

$$\boldsymbol{\eta}_j \sim \mathrm{MVN}(\boldsymbol{\alpha}, \boldsymbol{\Phi}) \tag{18}$$

$$E[\boldsymbol{\eta}_j] = \boldsymbol{\alpha} = \begin{bmatrix} \alpha_1 \\ \alpha_2 \\ \alpha_3 \end{bmatrix} \tag{19}$$

$$\mathrm{Var}[\boldsymbol{\eta}_j] = \boldsymbol{\Phi} = \begin{bmatrix} \phi_{11} & \phi_{12} & \phi_{13} \\ \phi_{21} & \phi_{22} & \phi_{23} \\ \phi_{31} & \phi_{32} & \phi_{33} \end{bmatrix}. \tag{20}$$

In the quadratic LCM, the factor loading matrix $\mathbf{\Lambda}$ is represented as

$$\mathbf{\Lambda} = \begin{bmatrix} \lambda_{11} & \lambda_{12} & \lambda_{13} \\ \vdots & \vdots & \vdots \\ \lambda_{51} & \lambda_{52} & \lambda_{53} \end{bmatrix} = \begin{bmatrix} 1 & 0 & 0 \\ 1 & 1 & 1 \\ 1 & 2 & 4 \\ 1 & 3 & 9 \\ 1 & 4 & 16 \end{bmatrix} \qquad (21)$$

with a size of $5 \times 3$. Fig 7 shows the path diagram of the quadratic LCM.

**Piecewise linear LCM.** To account for cases with a clear changing point and a sharp change in slope, we can apply the piecewise linear LCM. In the piecewise linear LCM, the location of the changing point is a critical issue. In this study, we evaluate models with changing points at 25%, 50%, and 75% of the exerted muscle strength. Compared to the quadratic LCM, the piecewise linear LCM has a different factor loading matrix $\mathbf{\Lambda}$. As a representative example, the factor loading matrix for the piecewise linear LCM with a changing point at 50%

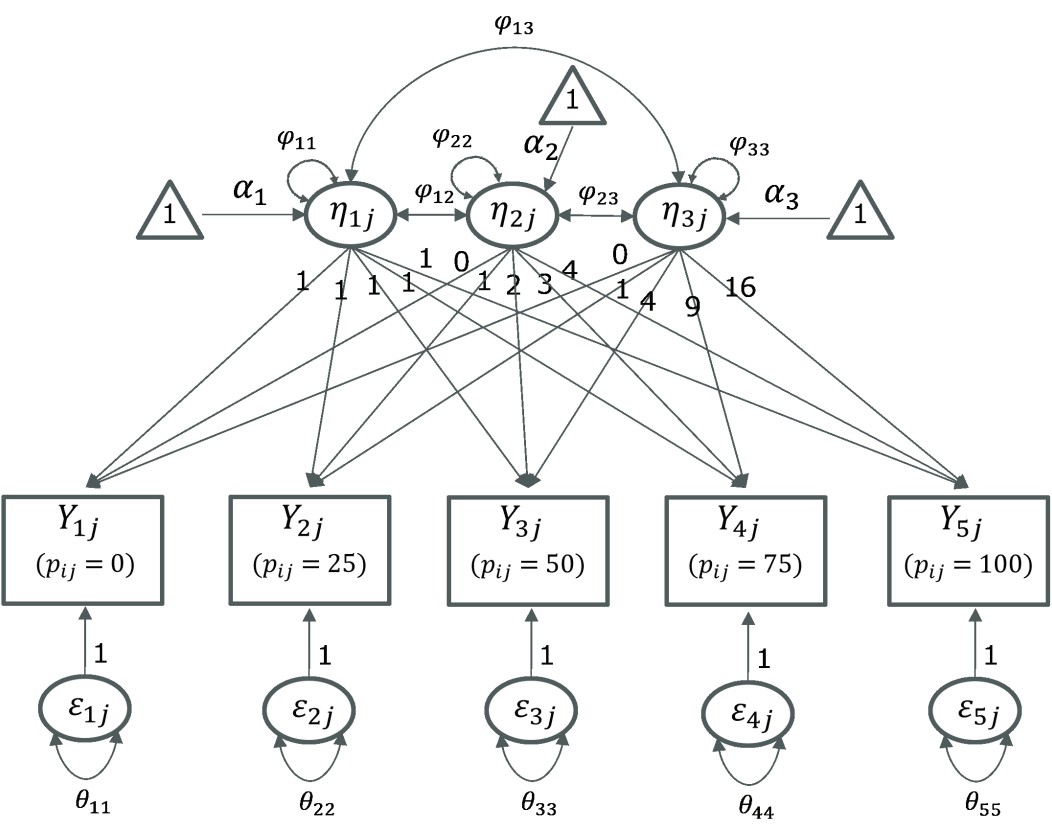

**Fig 7. Path diagram of the quadratic LCM.**

exerted muscle strength is shown as

$$\mathbf{\Lambda} = \begin{bmatrix} 1 & 0 & 0 \\ 1 & 1 & 0 \\ 1 & 2 & 0 \\ 1 & 2 & 1 \\ 1 & 2 & 2 \end{bmatrix}. \tag{22}$$

Fig 8 shows the piecewise linear LCM path diagram with a changing point at 50% exerted muscle strength.

**Multi-sample analysis and conditional LCM.** It is well known that the quantity and quality of muscle differ significantly between sexes [24,25]. Considering this, sex differences should be evident in the model. Therefore, we divided the data into two groups based on sex and used multi-sample analysis to fit the model to both datasets simultaneously to examine sex differences in the model. First, we divided the model by sex and freely estimated all parameters for each model. The $\chi^2$ statistic of this model served as the baseline fit. Next, we placed equality constraints on specific latent variables across sexes and estimated the parameters. This model's $\chi^2$ statistic was used as the comparison fit. If the comparison fit showed a decrease relative to the baseline fit, the latent variables with equality constraints negatively impacted the model fit, suggesting a sex difference in those latent variables. In this study, we

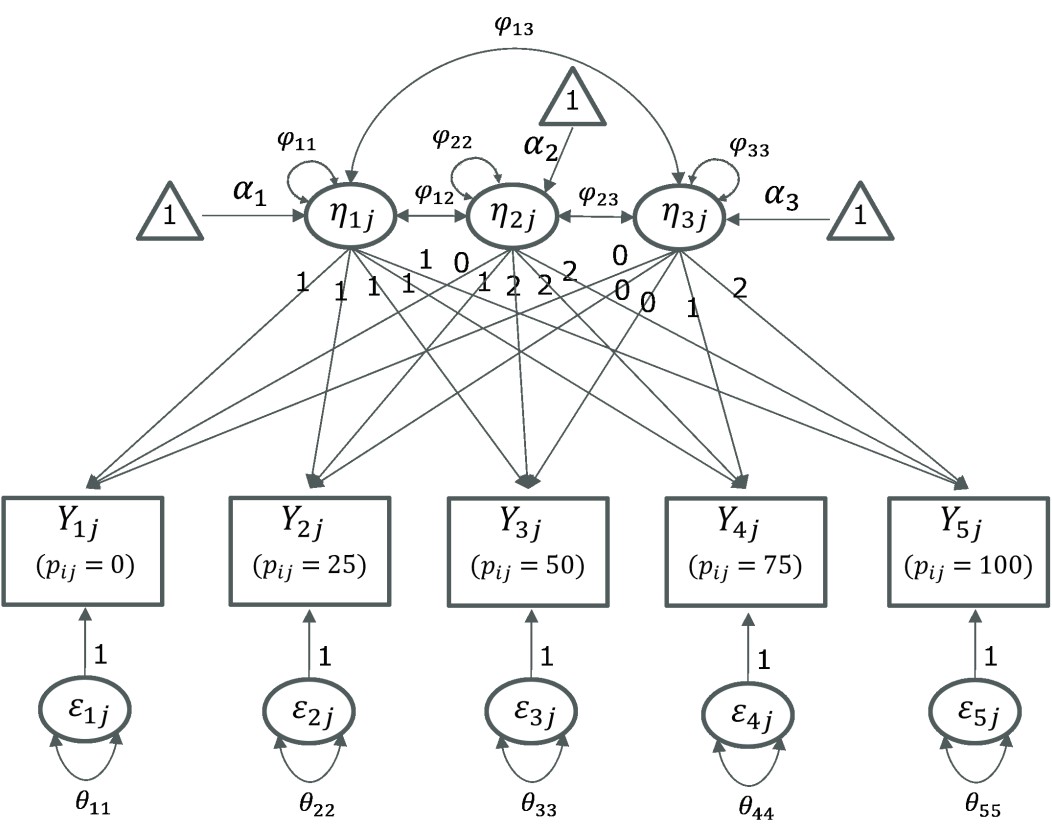

**Fig 8. Path diagram of the piecewise linear LCM.**

placed equality constraints on the variance-covariance matrix of the latent variables for both sexes and conducted hypothesis testing with the following null and alternative hypotheses:

- $H_0$: The models hold true with equality constraints over the groups (sex).
- $H_1$: Not $H_0$

Additionally, there is significant individual variability in muscle quantity and quality. For each sex, we modeled a conditional LCM that included explanatory variables related to muscle quantity and quality for the latent variables and evaluated the fit. In the case of the conditional linear LCM, let $\boldsymbol{x}_j$ be a vector of explanatory variables for each participant $j$ and $\boldsymbol{\delta}_j$ be an error vector, then $\boldsymbol{\eta}_j$ is expressed as

$$\boldsymbol{\eta}_j = \boldsymbol{\alpha} + \boldsymbol{\beta}\boldsymbol{x}_j^T + \boldsymbol{\delta}_j \tag{23}$$

$$\boldsymbol{\delta}_j \sim \mathrm{MVN}(\boldsymbol{0}, \boldsymbol{\Psi}). \tag{24}$$

The sizes of $\boldsymbol{\alpha}$ and $\boldsymbol{\beta}$ are determined by the size of the latent variable vector. For example, in the case of a quadratic LCM or a piecewise linear LCM, the sizes are $3 \times 1$.

## Results

### Examination of model structure

Piecewise regression can yield improper solutions when the error variances at each time point are estimated separately. Therefore, we estimated the parameters assuming the following error variance structure:

$$\mathrm{Var}[\boldsymbol{\epsilon_j}] = \boldsymbol{\Theta} = \begin{bmatrix} \theta_1 & 0 & 0 & 0 & 0 \\ 0 & \theta_2 & 0 & 0 & 0 \\ 0 & 0 & \theta_2 & 0 & 0 \\ 0 & 0 & 0 & \theta_3 & 0 \\ 0 & 0 & 0 & 0 & \theta_3 \end{bmatrix}. \tag{25}$$

Table 1 shows the fit indices for each model. Based on the fit criteria mentioned earlier, the quadratic LCM and the piecewise linear LCM with a changing point at 50% exerted muscle strength showed good values that met all fit criteria. On the other hand, the linear LCM had many indices that did not meet the fit criteria. The piecewise linear LCMs with changing points at 25% and 75% exerted muscle strength and also had a poor fit. Fig 9 shows the trajectories for the linear LCM, quadratic LCM, and piecewise linear LCM with a changing point at 50% exerted muscle strength. It can be seen that the trajectories of the quadratic LCM and the piecewise linear LCM, which had high goodness-of-fit, match. Additionally, there is a noticeable difference between the linear LCM and the nonlinear LCM, particularly for $p_{ij} \geq 50$. These observations suggest that the relationship between the observed values and the exerted muscle strength is likely nonlinear. Henceforth, the analysis will focus on the two models: the quadratic LCM and the piecewise linear LCM with a changing point at 50% exerted muscle strength.

### Multi-sample analysis

When fitted separately for males and females without imposing constraints across sexes, the fit indices for each model are shown in Table 2. Additionally, the list of parameters is shown

**Table 1. List of fit indices when fitting a linear LCM.**

| Model | $\chi^2$ | df | p-value | CFI | SRMR | RMSEA | TLI | AGFI |
|---|---|---|---|---|---|---|---|---|
| Linear LCM | 23.598 | 10 | 0.008 | 0.631 | 0.129 | 0.149 | 0.631 | 0.938 |
| Quadratic LCM | 6.663 | 6 | 0.353 | 0.995 | 0.030 | 0.042 | 0.992 | 0.974 |
| Piecewise LCM (25% changing point) | 42.132 | 8 | 0.000 | 0.775 | 0.109 | 0.262 | 0.719 | 0.874 |
| Piecewise LCM (50% changing point) | 10.436 | 8 | 0.235 | 0.984 | 0.051 | 0.070 | 0.979 | 0.969 |
| Piecewise LCM (75% changing point) | 52.419 | 8 | 0.000 | 0.707 | 0.100 | 0.299 | 0.634 | 0.842 |

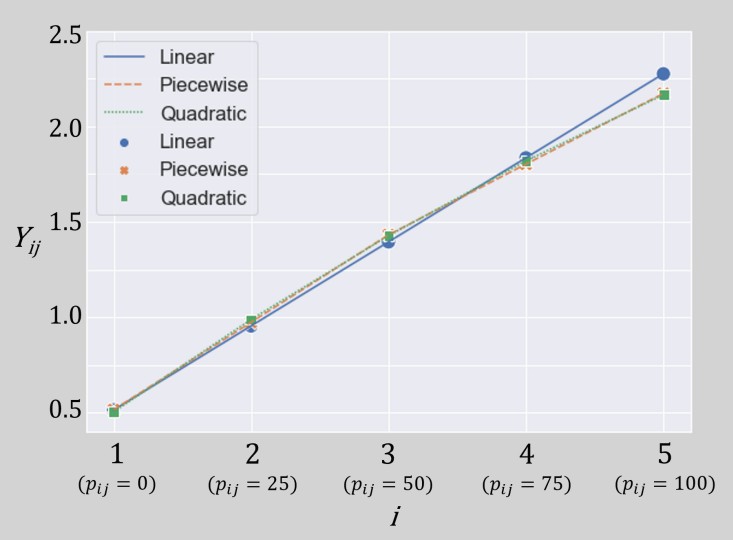

**Fig 9. Trajectories of the linear LCM, quadratic LCM, and piecewise linear LCM.**

**Table 2. List of fit indices when fitting the LCMs.**

| Model | $\chi^2$ | df | p-value | CFI | SRMR | RMSEA | TLI | AGFI |
|---|---|---|---|---|---|---|---|---|
| Quadratic (Male) | 15.865 | 8 | 0.044 | 0.914 | 0.106 | 0.178 | 0.893 | 0.913 |
| Quadratic (Female) | 5.607 | 8 | 0.691 | 1.000 | 0.045 | 0.000 | 1.044 | 0.970 |
| Piecewise (Male) | 5.076 | 6 | 0.534 | 1.000 | 0.080 | 0.000 | 1.016 | 0.962 |
| Piecewise (Female) | 8.964 | 6 | 0.175 | 0.955 | 0.094 | 0.126 | 0.926 | 0.935 |

in Tables 3 and 4. The parameters were estimated using the maximum likelihood method, and optimization was performed using the Newton-Raphson method [26]. To avoid improper solutions, the error variance structure was assumed as follows:

$$\text{Var}[\boldsymbol{\epsilon_j}] = \boldsymbol{\Theta} = \begin{bmatrix} \theta_1 & 0 & 0 & 0 & 0 \\ 0 & \theta_2 & 0 & 0 & 0 \\ 0 & 0 & \theta_2 & 0 & 0 \\ 0 & 0 & 0 & \theta_3 & 0 \\ 0 & 0 & 0 & 0 & \theta_3 \end{bmatrix}. \tag{26}$$

In some models, the covariance matrix of the latent variables was found to be non-positive definite. This could suggest that the model structure is inappropriate for the data. However,

**Table 3. List of quadratic LCM parameters estimated without equality constraints between sexes.**

| Quadratic | Males | | | | Females | | | |
|---|---|---|---|---|---|---|---|---|
| | Estimate | Std.Err | z-value | P(> \|z\|) | Estimate | Std.Err | z-value | P(> \|z\|) |
| $\alpha_1$ | 0.537 | 0.045 | 11.991 | 0.000*** | 0.471 | 0.039 | 11.955 | 0.000*** |
| $\alpha_2$ | 0.299 | 0.078 | 3.857 | 0.000*** | 0.723 | 0.075 | 9.629 | 0.000*** |
| $\alpha_3$ | 0.019 | 0.025 | 0.752 | 0.452 | −0.068 | 0.021 | −3.245 | 0.001** |
| $\theta_1$ | 0.013 | 0.027 | 0.468 | 0.640 | 0.029 | 0.025 | 1.167 | 0.243 |
| $\theta_2$ | 0.052 | 0.013 | 4.045 | 0.000*** | 0.042 | 0.011 | 3.746 | 0.000*** |
| $\theta_3$ | 0.052 | 0.013 | 4.045 | 0.000*** | 0.042 | 0.011 | 3.746 | 0.000*** |
| $\theta_4$ | 0.054 | 0.017 | 3.179 | 0.001** | 0.221 | 0.058 | 3.800 | 0.000*** |
| $\theta_5$ | 0.054 | 0.017 | 3.179 | 0.001** | 0.221 | 0.058 | 3.800 | 0.000*** |
| $\phi_{11}$ | 0.050 | 0.030 | 1.675 | 0.097· | 0.020 | 0.025 | 0.805 | 0.421 |
| $\phi_{22}$ | 0.147 | 0.051 | 2.869 | 0.004** | 0.115 | 0.050 | 2.321 | 0.020* |
| $\phi_{33}$ | 0.017 | 0.005 | 3.329 | 0.001** | 0.009 | 0.004 | 2.341 | 0.019* |
| $\phi_{12}$ | −0.048 | 0.031 | −1.519 | 0.129 | 0.030 | 0.027 | 1.108 | 0.268 |
| $\phi_{13}$ | 0.016 | 0.008 | 1.923 | 0.055· | −0.009 | 0.006 | −1.488 | 0.137 |
| $\phi_{23}$ | −0.042 | 0.015 | −2.817 | 0.005** | −0.025 | 0.012 | −2.029 | 0.042* |

Levels of significance are indicated by asterisks: $*** p<0.001$, $** p<0.01$, $* p<0.05$, $· p < 0.1$.

**Table 4. List of piecewise linear LCM parameters estimated without equality constraints between sexes.**

| Piecewise | Males | | | | Females | | | |
|---|---|---|---|---|---|---|---|---|
| | Estimate | Std.Err | z-value | P(> \|z\|) | Estimate | Std.Err | z-value | P(> \|z\|) |
| $\alpha_1$ | 0.529 | 0.042 | 12.692 | 0.000*** | 0.491 | 0.040 | 12.160 | 0.000*** |
| $\alpha_2$ | 0.338 | 0.047 | 7.160 | 0.000*** | 0.585 | 0.047 | 12.543 | 0.000*** |
| $\alpha_3$ | 0.414 | 0.101 | 4.082 | 0.000*** | 0.326 | 0.077 | 4.259 | 0.000*** |
| $\theta_1$ | 0.037 | 0.021 | 1.771 | 0.077· | 0.038 | 0.021 | 1.768 | 0.077· |
| $\theta_2$ | 0.037 | 0.010 | 3.729 | 0.000*** | 0.057 | 0.015 | 3.854 | 0.000*** |
| $\theta_3$ | 0.037 | 0.010 | 3.729 | 0.000*** | 0.057 | 0.015 | 3.854 | 0.000*** |
| $\theta_4$ | 0.050 | 0.014 | 3.563 | 0.000*** | 0.223 | 0.058 | 3.836 | 0.000*** |
| $\theta_5$ | 0.050 | 0.014 | 3.563 | 0.000*** | 0.223 | 0.058 | 3.836 | 0.000*** |
| $\phi_{11}$ | 0.023 | 0.020 | 1.142 | 0.253 | 0.017 | 0.021 | 0.810 | 0.418 |
| $\phi_{22}$ | 0.051 | 0.018 | 2.804 | 0.005** | 0.045 | 0.019 | 2.409 | 0.016* |
| $\phi_{33}$ | 0.298 | 0.081 | 3.685 | 0.000*** | 0.121 | 0.048 | 2.530 | 0.011* |
| $\phi_{12}$ | −0.007 | 0.015 | −0.464 | 0.643 | 0.019 | 0.015 | 1.298 | 0.194 |
| $\phi_{13}$ | 0.035 | 0.025 | 1.414 | 0.157 | −0.035 | 0.018 | −1.917 | 0.055· |
| $\phi_{23}$ | −0.007 | 0.027 | −0.275 | 0.783 | 0.017 | 0.020 | 0.836 | 0.403 |

Levels of significance are indicated by asterisks: $*** p<0.001$, $** p<0.01$, $* p<0.05$, $· p < 0.1$.

upon checking the eigenvalues, although there were negative eigenvalues, they were sufficiently small. The error variance $\theta_i$ tends to increase with the increase in exerted muscle strength. This is likely due to the experimental noise associated with controlling the exerted muscle strength increasing with higher exerted muscle strength.

The z-test results indicated that $\alpha_3$ was not significant in the quadratic LCM for males. On the other hand, in the piecewise linear LCM, all factors representing the pattern of changes, including $\alpha_3$, were found to be significant. Additionally, while the model fit indices for the quadratic LCM for males did not meet the criteria, the piecewise linear LCM showed a good fit. This suggests that for males, the relationship might be better represented by the piecewise linear LCM, indicating a sharper change in the rate of increase. For females, all factors representing the pattern of changes in both the quadratic LCM and the piecewise linear LCM were found to be significant. However, the quadratic LCM had a better fit. This suggests that for females, unlike males, the relationship might involve a more gradual change in the

rate of increase. Fig 10 shows the trajectories of the piecewise linear LCM for males and the quadratic LCM for females.

In the quadratic LCM, $\alpha_3$ was positive for males and negative for females. For both sexes, the observed values increased monotonically with increasing exerted muscle strength. However, for males, the rate of increase in observed values was greater with increasing exerted muscle strength, while for females, the rate of increase was smaller. However, regarding the trajectory of the piecewise linear LCM for males, there was no significant change in the slope before and after the turning point, resulting in a trajectory that closely resembled that of a linear LCM. This trend was also suggested by the relationship between $\alpha_2$ and $\alpha_3$ in the piecewise linear LCM. The results of the significance tests for these differences are shown in Tables 5 and 6. In the quadratic LCM, $H_0$ was rejected when equality constraints were placed on $\alpha^2$ and $\alpha^3$. This indicates that there was no significant sex difference in the intercept, but there were significant sex differences in the linear and quadratic slopes. In the piecewise linear LCM, $H_0$ was rejected when an equality constraint was placed on $\alpha^2$. This suggests no significant sex difference in the intercept for the piecewise linear LCM either. Additionally, there was no significant sex difference in the slope at high-exerted muscle strength. However, there was a significant sex difference in the slope at low-exerted muscle strength.

Summarizing the model fit for both sexes, males tend to show a sharp increase in the rate of change of observed values around 50% exerted muscle strength. Females tend to show a gradual decrease in the rate of change of observed values. Notably, up to 50% exerted muscle strength, the rate of change in observed values differs between males and females.

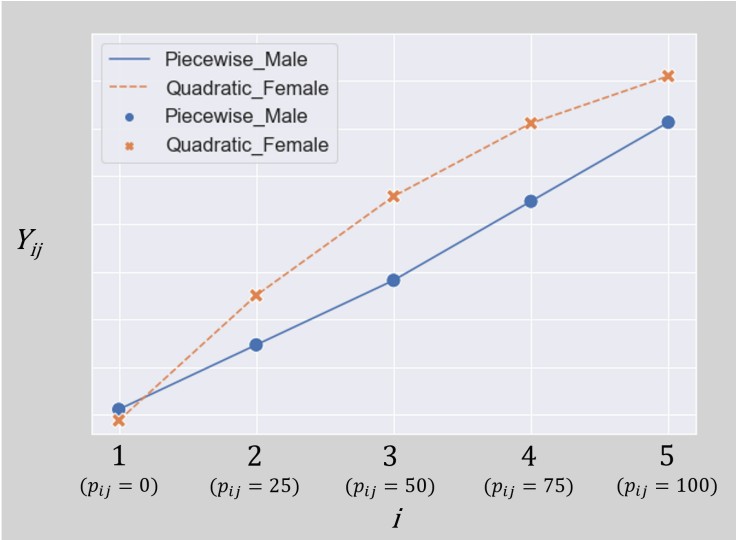

**Fig 10. Comparison of the trajectories of the piecewise linear LCM for males and the quadratic LCM for females.**

**Table 5. Chi-square difference test for quadratic LCM with equality constraints.**

| Latent Variable with Equality Constraint | Chi-square Difference | Difference in df | p-value |
|---|---|---|---|
| $\alpha^1$ | 1.248 | 1 | 0.263 |
| $\alpha^2$ | 12.663 | 1 | <0.001 |
| $\alpha^3$ | 6.037 | 1 | 0.014 |

**Table 6. Chi-square difference test for piecewise linear LCM with equality constraints.**

| Latent Variable with Equality Constraint | Chi-square Difference | Difference in df | p-value |
|---|---|---|---|
| $\alpha^1$ | 0.313 | 1 | 0.575 |
| $\alpha^2$ | 11.947 | 1 | <0.001 |
| $\alpha^3$ | 0.445 | 1 | 0.504 |

## Conditional LCM

In the quadratic LCM, there was no significant sex difference in the intercept, but there were significant sex differences in the linear and quadratic slopes. Considering that muscle properties differ between sexes, it is likely that the linear and quadratic slopes also vary depending on muscle properties. Additionally, the quadratic LCM showed a high goodness-of-fit for females. Therefore, we hypothesized that in the quadratic LCM for females, individual differences in the linear and quadratic slopes are influenced by muscle properties.

In the piecewise linear LCM, there was no significant sex difference in the intercept or the slope from 50% to 100% exerted muscle strength, but there was a significant sex difference in the slope from 0% to 50%. Considering the differences in muscle properties between sexes, it is likely that the slope from 0% to 50% also varies depending on muscle properties. The piecewise linear LCM showed a high goodness-of-fit for males. Therefore, we hypothesized that in the piecewise linear LCM for males, individual differences in the slope from 0% to 50% are influenced by muscle properties.

Thus, we divided the data by sex and examined the significance and goodness-of-fit of the explanatory variables added to the conditional LCM for the quadratic LCM for females and the piecewise linear LCM for males. The goodness-of-fit of these conditional LCMs and the results of the z-tests for each explanatory variable are shown in Tables 7 and 8. In the quadratic LCM for females, only the maximum torque value showed a high goodness-of-fit and significance as an explanatory variable. This maximum torque value was significant only for the linear slope, suggesting that participants with higher torque values tend to have a larger linear slope.

In the piecewise linear LCM for males, both body fat percentage and ankle circumference were significant explanatory variables for the slope from 0% to 50%. Both explanatory variables suggested that participants with higher values tend to have a smaller slope from 0% to 50%.

**Table 7. Parameter estimates for the quadratic LCM for females.**

| Explanatory variables | | Estimate | $P(>|z|)$ | $\chi^2$ | CFI | SRMR | RMSEA | TLI | AGFI |
|---|---|---|---|---|---|---|---|---|---|
| $x_j$=Height | $\beta_2$ | −0.006 | 0.959 | 9.754 | 1.000 | 0.077 | 0.000 | 1.059 | 0.957 |
| | $\beta_3$ | 0.004 | 0.915 | | | | | | |
| $x_j$=Fat | $\beta_2$ | −0.007 | 0.927 | 14.648 | 0.976 | 0.081 | 0.064 | 0.972 | 0.934 |
| | $\beta_3$ | 0.015 | 0.500 | | | | | | |
| $x_j$=Ankle | $\beta_2$ | 0.066 | 0.445 | 15.005 | 0.971 | 0.080 | 0.070 | 0.966 | 0.935 |
| | $\beta_3$ | −0.026 | 0.282 | | | | | | |
| $x_j$=Grip | $\beta_2$ | 0.039 | 0.722 | 14.021 | 0.985 | 0.085 | 0.050 | 0.982 | 0.938 |
| | $\beta_3$ | −0.003 | 0.934 | | | | | | |
| $x_j$=Torque | $\beta_2$ | 0.157 | 0.036* | 10.723 | 1.000 | 0.092 | 0.000 | 1.034 | 0.957 |
| | $\beta_3$ | −0.009 | 0.694 | | | | | | |

Levels of significance are indicated by asterisks: $***$ $p$<0.001, $**$ $p$<0.01, $*$ $p$<0.05, $\cdot$ $p < 0.1$.

**Table 8. Parameter estimates for the piecewise linear LCM for males.**

| Explanatory variables | | Estimate | $P(>\lvert z\rvert)$ | $\chi^2$ | CFI | SRMR | RMSEA | TLI | AGFI |
|---|---|---|---|---|---|---|---|---|---|
| $x_j$ =Height | $\beta_2$ | 0.072 | 0.202 | 9.929 | 1.000 | 0.086 | 0.000 | 1.047 | 0.977 |
| $x_j$ =Fat | $\beta_2$ | −0.086 | 0.057· | 19.564 | 0.945 | 0.126 | 0.113 | 0.942 | 0.938 |
| $x_j$ =Ankle | $\beta_2$ | −0.077 | 0.062· | 15.280 | 0.987 | 0.091 | 0.054 | 0.986 | 0.946 |
| $x_j$ =Grip | $\beta_2$ | 0.062 | 0.504 | 15.280 | 1.000 | 0.092 | 0.000 | 1.061 | 0.960 |
| $x_j$ =Torque | $\beta_2$ | 0.056 | 0.442 | 10.968 | 1.000 | 0.093 | 0.000 | 1.035 | 0.955 |

Levels of significance are indicated by asterisks: $*** p<0.001$, $** p<0.01$, $* p<0.05$, $\cdot p < 0.1$.

The results of the piecewise linear LCM for males indicated that individuals with higher body fat percentages and thicker ankles have a lower rate of increase in signal. Based on the hypothesis that the signal is emitted from the tendon and captured by the sensor on the skin surface, this is consistent with the hypothesis that the greater the fat thickness between the tendon and the skin surface, the more the signal is attenuated. This suggests that the biological signals being captured likely originate from the tendon.

## Discussion

In this study, we hypothesized that the biological signals generated by muscle contraction are transmitted to the tendon and observed as MTG. A typical signal generated by muscle contraction is MMG [6], which results from lateral expansion of the muscle during contraction. This expansion generates a pressure wave that propagates through tissue and causes body surface vibration. Because MMG requires sensors to be placed on the muscle belly, the vibrations transmitted to the body surface can vary depending on factors such as muscle quality and fat amount, potentially not faithfully representing the characteristics of muscle contraction. If this hypothesis is correct, the relationship between exerted muscle strength and MTG should reflect the physiological characteristics of muscle contraction. In the following sections, we examine the physiological mechanisms underlying this relationship, including sex differences, and discuss how they align with our modeling results. Based on this, we will discuss the correlation between the modeling results and sex differences.

### Relationship between exerted muscle strength and muscle contraction, and sex differences

**Properties of muscle fibers and motor units.** Muscle strength increases as more muscle fibers contract, triggered by action potentials from motor neurons [27]. This process, known as motor unit firing, activates only the fibers innervated by the firing neuron.

A motor unit consists of a motor neuron and its connected muscle fibers, typically ranging from 5 to 2000 [28]. The size and properties of motor units can be broadly classified into the following three types:

- Slow (S) type

   They are used for activities requiring sustained low-force output, such as maintaining posture or endurance. The number of muscle fibers connected to a single motor neuron is small. The connected muscle fibers are predominantly Type I fibers, also known as slow-twitch fibers. These fibers have a high content of mitochondria and myoglobin, making them highly efficient in oxidative metabolism, using oxygen to generate energy. A single

unit's firing frequency is typically 10-30 Hz, providing a sustained and stable firing pattern. As a result, muscle contractions are also stable over time.

- Fast Fatigable (FF) type

  They are used for explosive movements such as sprinting or jumping. The number of muscle fibers connected to a single motor neuron is large. The connected muscle fibers are predominantly Type IIx/IIb fibers, also known as fast-twitch fibers. These fibers excel in glycolytic energy metabolism, rapidly producing energy. They have a contraction speed about 4.1 times faster than Type I fibers, allowing for the generation of large forces. The firing frequency for a single unit is typically in the range of 30-100 Hz. Due to the high firing frequency and the need for very strong contractions over a short period, the firing pattern tends to be unstable. Consequently, muscle contractions per unit time are also unstable.

- Fast Fatigue-Resistant (FR) type

  They are used for weight training or typical daily activities. These fibers have intermediate properties between S and FF types. The connected muscle fibers are predominantly Type IIa fibers.

It should be noted that while we have categorized motor units into three types for explanation purposes, many studies have pointed out that the properties of muscle fibers are continuous and strict classification is difficult [29]. S-type motor units have a small exerted muscle strength per muscle fiber and are inefficient. Therefore, a large number of contracting muscle fibers would be required to increase exerted muscle strength. On the other hand, FF-type motor units have a large exerted muscle strength per muscle fiber and are efficient. Thus, fewer contracting muscle fibers would be required to increase exerted muscle strength.

Motor units are recruited according to the size principle: S-type (Type I) units first, followed by FR (Type IIa) and FF (Type IIx/IIb) units as force demands increase.

**Composition ratio of muscle fibers and sex differences.** The Achilles tendon connects to both the gastrocnemius and the soleus muscles, which differ in motor unit composition [30]. The gastrocnemius has more FF-type units, while the larger soleus (2× volume) predominantly contains S-type units. Consequently, Type I fibers are more abundant in the region associated with the Achilles tendon. While the total number of muscle fibers is similar between sexes, their composition differs [25,31]. Females typically have more Type I and fewer Type IIa fibers, whereas males show the opposite trend. Type IIx/IIb proportions are generally similar. Below is a summary of the relationship between the increase in exerted muscle strength and the number of muscle fibers recruited, categorized by sex.

- Females:
  - At low exertion: More Type I fibers → lower per-fiber force → more fibers recruited.
  - At high exertion: Most fibers already recruited → limited Type II reserve → plateau in recruitment.
- Males:
  - At low exertion: Fewer Type I fibers → lower recruitment rate.
  - At high exertion: Larger Type II reserve → continued recruitment.

These differences may influence MTG signal patterns as exerted muscle strength increases.

## Consistency between modeling results and muscle contraction mechanism

MTG increased monotonically with exerted muscle strength, consistent with previous findings. Among all tested models, the quadratic LCM best fit the female data, and a piecewise

linear LCM with a change point at 50% fit the male data. These models showed high goodness of fit, suggesting a gradual increase in females and an abrupt increase in males.

The piecewise linear model showed a significantly steeper slope in females at low exertion levels. At high exertion, males showed a greater slope, though the difference was not statistically significant. Intercepts also showed no significant sex difference, indicating similar initial MTG values. Here are the points of consistency between the modeling results and the muscle contraction mechanisms, along with the hypotheses derived from them.

- Model structure differences:
  Females, with fewer Type IIa fibers, likely experience earlier fiber depletion as exertion increases, resulting in a gradually decreasing slope—well modeled by a quadratic LCM. Males, with more Type IIa fibers, sustain recruitment over a broader range, fitting a piecewise linear pattern with minimal slope change.
- Low exertion slope differences:
  Females recruit more Type I fibers at low exertion, resulting in steeper slopes. In males, earlier recruitment of Type IIa fibers results in a slower increase in contracting fibers. These patterns align with section Multi-sample analysis.
- High exertion slope differences:
  Females likely deplete Type IIa fibers earlier, reducing slope at high exertion. Males maintain a larger reserve of Type IIa fibers, sustaining the recruitment rate. This is consistent with the trends in Fig 10.

These findings suggest that MTG may reflect the number of contracting fibers. The sex differences at maximal exertion likely reflect greater Type I fiber recruitment in females. If MTG amplification patterns correspond to fiber recruitment stages, MTG could offer a noninvasive method to estimate fiber composition. Detecting slope changes may help identify the transition from Type I to Type IIa recruitment, aiding in personalized training strategies.

## Conclusions

In this study, we focused on MTG as a method for quantifying muscle activity and attempted to identify the detailed relationship between exerted muscle strength and MTG using LCM. Experiments were conducted on subjects, and MTG data at each exerted muscle strength were obtained using piezoelectric film sensors. A longitudinal data structure was formed by processing the obtained data from 62 individuals at five different points in time. Various LCM structures were applied to this longitudinal data to identify the model that best represents the data structure. As a result, it was found that the models fitting both males and females were nonlinear, but their shapes differed. It was also suggested that there was a significant sex difference in the rate of increase in MTG at low levels of exerted muscle strength. Using conditional LCM for each sex, the significance and goodness-of-fit of explanatory variables added to the latent variables were examined. The results showed that significant explanatory variables existed for both sexes. In particular, the significant explanatory variables and their coefficients for the male latent variables supported the hypothesis that MTG is generated from the Achilles tendon.

The discussion examined the correlation between these modeling results and physiological knowledge. The relationship between exerted muscle strength and MTG was found to correlate with the relationship between exerted muscle strength and the total number of contracting muscle fibers. Notably, the sex differences observed in the modeling results were consistent with those in muscle fiber properties. This discussion result suggests that fine vibrations

cause the generation of MTG due to muscle fiber contractions, and these fine vibrations are transmitted to the tendons connected to the muscles, supporting the hypothesis that MTG is observed from these tendons.

In the future, we will focus on verifying this statistical hypothesis through further research. By conducting experiments with an increased number of time points, we will examine whether the location of the breakpoint changes depending on muscle composition. We will consider introducing methods such as intensive longitudinal data analysis to address the challenges associated with analyzing an increased number of time points. Additionally, we will conduct physiological experiments to confirm that the hypotheses proposed in this study actually occur within the body. Confirming this will not only contribute to understanding human mechanisms but also demonstrate that understanding phenomena through statistical modeling of biological data is effective in elucidating human mechanisms.

## Acknowledgments

We extend our deepest gratitude to Yuichi Motohisa and Nozomi Matsunaga of the Nagai Cardiovascular Internal Medicine Clinic and Chiharu Fujisawa from Shikoku Medical School for their invaluable cooperation and support. The expertise and dedication of these individuals were instrumental in the successful completion of our experiments and fundamental to achieving the profound insights gained from this research. We would also like to express our appreciation to Atsushi Naito, Naoki Kawara, Yutaka Takamaru, and Risako Yamashita from Murata Manufacturing Co., Ltd. for their assistance and expertise, which contributed significantly to our research.

## Author contributions

**Data curation:** Tatsuhiko Matsumoto.

**Investigation:** Tatsuhiko Matsumoto.

**Methodology:** Tatsuhiko Matsumoto, Yutaka Kano.

**Project administration:** Tatsuhiko Matsumoto, Yutaka Kano.

**Resources:** Tatsuhiko Matsumoto.

**Software:** Tatsuhiko Matsumoto.

**Supervision:** Yutaka Kano.

**Validation:** Tatsuhiko Matsumoto.

**Visualization:** Tatsuhiko Matsumoto.

**Writing – original draft:** Tatsuhiko Matsumoto.

**Writing – review & editing:** Tatsuhiko Matsumoto, Yutaka Kano.

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
