## [Decision Letter · Decision Letter 0]

14 Apr 2025

PONE-D-24-45657Increase trajectories of tendon micro vibration intensity during ankle plantar flexion: A longitudinal data analysis using latent curve modelsPLOS ONE

Dear Dr. Matsumoto,

Thank you for submitting your manuscript to PLOS ONE. After careful consideration, we feel that it has merit but does not fully meet PLOS ONE’s publication criteria as it currently stands. Therefore, we invite you to submit a revised version of the manuscript that addresses the points raised during the review process.

**ACADEMIC EDITOR: **The submission has scientific merits, however, requires a major revision to address the concerns raised by the reviewers. 

We look forward to receiving your revised manuscript.

Kind regards,

Yusuf Oloruntoyin Ayipo, Ph.D

Academic Editor

PLOS ONE

Journal Requirements:

2. In this instance it seems there may be acceptable restrictions in place that prevent the public sharing of your minimal data. However, in line with our goal of ensuring long-term data availability to all interested researchers, PLOS’ Data Policy states that authors cannot be the sole named individuals responsible for ensuring data access (http://journals.plos.org/plosone/s/data-availability#loc-acceptable-data-sharing-methods).

Before we proceed with your manuscript, please

 also provide non-author contact information (phone/email/hyperlink) for a data access committee, ethics committee, or other institutional body to which data requests may be sent. If no institutional body is available to respond to requests for your minimal data, please consider if there any institutional representatives who did not collaborate in the study, and are not listed as authors on the manuscript, who would be able to hold the data and respond to external requests for data access? If so, please provide their contact information (i.e., email address). Please also provide details on how you will ensure persistent or long-term data storage and availability.

Additional Editor Comments:

The submission has scientific merits, however, requires a major revision to address the concerns raised by the reviewers.

Reviewers' comments:

Reviewer's Responses to Questions

**Comments to the Author**

1. Is the manuscript technically sound, and do the data support the conclusions?

Reviewer #1: Yes

Reviewer #2: Yes

Reviewer #3: Yes

Reviewer #4: Yes

2. Has the statistical analysis been performed appropriately and rigorously? 

Reviewer #1: Yes

Reviewer #2: Yes

Reviewer #3: Yes

Reviewer #4: Yes

3. Have the authors made all data underlying the findings in their manuscript fully available?

Reviewer #1: Yes

Reviewer #2: Yes

Reviewer #3: Yes

Reviewer #4: Yes

4. Is the manuscript presented in an intelligible fashion and written in standard English?

Reviewer #1: Yes

Reviewer #2: Yes

Reviewer #3: Yes

Reviewer #4: Yes

5. Review Comments to the Author

Reviewer #1: This manuscript requires a significant amount of improvement in:

Provide number of time points and intervals at which measurements are taken,

using the time metric that best reflects the time of inclusion in the study

(typically time from enrollment, or calendar time in studies that involve long

enrollment times).

Highlight the differences between the time of first

measurements and follow-up times

Describe patterns of missing values across variables at each time point and

across time points.

Describe the participants that have missing data for some of the measurements

Estimate the probability of drop-out after inclusion, taking appropriately into

account the reasons for drop-out

Visualize the association between each explanatory variable with the structural

variables at baseline

Summarize changes and variability of variables within subjects

Describe numerically or graphically the longitudinal trends of the time-varying

variables

Reviewer #2: he statistical analysis in this study is well-designed and rigorously executed. The use of latent curve models (LCM) is appropriate for analyzing longitudinal data on tendon micro-vibration intensity across varying levels of exerted muscle strength. The authors thoroughly evaluated multiple LCM structures (linear, quadratic, piecewise linear) and selected the best-fitting models based on established goodness-of-fit criteria (e.g., CFI, RMSEA, SRMR). The inclusion of sex-specific analyses and multi-sample comparisons further strengthens the study, as it accounts for known physiological differences between males and females.

Strengths:

Robust Methodology: The preprocessing steps (noise removal, peak detection) and model validation (fit indices, chi-square tests) are clearly described and justified.

Physiological Consistency: The results align with existing knowledge of muscle fiber composition and motor unit recruitment, lending credibility to the findings.

Transparency: The authors acknowledge limitations, such as model fit challenges for males and restricted data availability, which enhances the study's reliability.

Suggestions for Improvement:

Model Fit Justification: While the piecewise linear LCM fits males well, the borderline fit indices for the quadratic LCM (RMSEA = 0.178) could be discussed further—perhaps exploring alternative nonlinear models.

Data Accessibility: Although ethical restrictions are understandable, providing a de-identified dataset or simulation code (if feasible) would improve reproducibility.

Future Work: The hypothesis that MTG reflects muscle fiber contractions could be strengthened by correlating findings with direct physiological measurements (e.g., EMG/MMG) in follow-up studies.

Ethical Considerations:

The study complies with ethical guidelines (Helsinki Declaration, institutional approval) and clearly outlines participant consent and exclusion criteria. No dual publication or ethical concerns are evident.

Reviewer #3: Good job by the authors. The corrections suggested in the attachment should be carefully considered. I did observe that manuscript is available on ResearchGate. I don't know if this violates the dual publication requirement, and so I let the editor(s) know.

Reviewer #4: This manuscript introduces a novel and robust method for measuring muscle activity through mechanotendography (MTG) that is combined with a latent curve model. The research methods are detailed and also provide solid support for the conclusion. That said, a few minor changes are recommended. The present Data Availability statement does not totally comply with PLOS ONE's open data policies. The authors should consider sharing an unidentified version of the dataset in a public database, or provide a clear explanation if ethical considerations prevents this. Given that the writing is clear, in my estimation, the introduction and discussion might need a few modifications for simplicity and conciseness. Ultimately, this is a well-conducted study that provides valuable profound knowledge to it's readers and I propose that it be published with minor adjustments focusing on data accessibility and presentation clarity.

6. PLOS authors have the option to publish the peer review history of their article (what does this mean?). If published, this will include your full peer review and any attached files.

Reviewer #1: No

Reviewer #2: No

Reviewer #3: No

Reviewer #4: No

---

## [Author Response · Author response to Decision Letter 1]

28 May 2025

Thank you for the constructive comments. We appreciate the kindness and efforts of the reviewers and the Academic Editor very much.

We have revised the manuscript following all the comments.

In particular,

- The Introduction and Discussion sections were improved to achieve simplicity and conciseness as they suggested.

- The Data Availability Statement was updated in the submission system.

An anonymized and processed dataset has been published on Figshare (DOI provided), and institutional contact information has been included to ensure long-term data accessibility.

We have thoroughly addressed all other points raised by the reviewers and explained the changes in the uploaded "Response to Reviewers"

document. We created some graphs to address their comments and they are given in the Response to Reviewers document. We feel that we should probably avoid enlarging the manuscript so the additional graphs were not included in it. However, we would be happy to incorporate them into the manuscript if the Editor deems it appropriate.

---

## [Decision Letter · Decision Letter 1]

22 Jun 2025

Increase trajectories of tendon micro vibration intensity during ankle plantar flexion: A longitudinal data analysis using latent curve models

PONE-D-24-45657R1

Dear Dr. Matsumoto,

We’re pleased to inform you that your manuscript has been judged scientifically suitable for publication and will be formally accepted for publication once it meets all outstanding technical requirements.

Kind regards,

Yusuf Oloruntoyin Ayipo, Ph.D

Academic Editor

PLOS ONE

Additional Editor Comments (optional):

The submission meets the level of scientific rigour required for publication in this title and all the concerns raised by the respective reviewers have been addressed satisfactorily. I hereby recommend the manuscript for publication in the current version.

Reviewers' comments:

Reviewer's Responses to Questions

**Comments to the Author**

1. If the authors have adequately addressed your comments raised in a previous round of review and you feel that this manuscript is now acceptable for publication, you may indicate that here to bypass the “Comments to the Author” section, enter your conflict of interest statement in the “Confidential to Editor” section, and submit your "Accept" recommendation.

Reviewer #3: All comments have been addressed

Reviewer #4: All comments have been addressed

2. Is the manuscript technically sound, and do the data support the conclusions?

Reviewer #3: Yes

Reviewer #4: Yes

3. Has the statistical analysis been performed appropriately and rigorously? 

Reviewer #3: Yes

Reviewer #4: Yes

4. Have the authors made all data underlying the findings in their manuscript fully available?

Reviewer #3: Yes

Reviewer #4: Yes

5. Is the manuscript presented in an intelligible fashion and written in standard English?

Reviewer #3: Yes

Reviewer #4: Yes

6. Review Comments to the Author

Reviewer #3: (No Response)

Reviewer #4: This publication proposes an original and practical method for assessing muscle activity using mechanotendography (MTG), which has the potential to have a substantial impact in both clinical and daily circumstances. The rationale for MTG as an alternative to EMG and MMG has been demonstrated, and the methodical utilization of latent curve models is ideal for modeling longitudinal muscle response data. The outcomes, most especially the sex-specific model differences, are intriguing and merit further study. To improve the writing, the authors should discuss the physiological rationale for sex variations and clarify sensor placement repeatability. In the end, this is an organized and significant contribution to non-invasive muscle monitoring methods.

7. PLOS authors have the option to publish the peer review history of their article (what does this mean?). If published, this will include your full peer review and any attached files.

Reviewer #3: No

Reviewer #4: No

---

## [Editor Report · Acceptance letter]

PONE-D-24-45657R1

PLOS ONE

Dear Dr. Matsumoto,

I'm pleased to inform you that your manuscript has been deemed suitable for publication in PLOS ONE. Congratulations! Your manuscript is now being handed over to our production team.

Kind regards,

on behalf of

Dr. Yusuf Oloruntoyin Ayipo

Academic Editor

PLOS ONE